# Safety Profile of COVID-19 Vaccines among Healthcare Workers in Poland

**DOI:** 10.3390/vaccines10030434

**Published:** 2022-03-12

**Authors:** Anna Paczkowska, Karolina Hoffmann, Michał Michalak, Anna Hans-Wytrychowska, Wiesław Bryl, Dorota Kopciuch, Tomasz Zaprutko, Piotr Ratajczak, Elżbieta Nowakowska, Krzysztof Kus

**Affiliations:** 1Department of Pharmacoeconomics and Social Pharmacy, Poznan University of Medical Sciences, Rokietnicka 7 Street, 60-806 Poznań, Poland; dorota.koligat@gmail.com (D.K.); tomekzaprutko@ump.edu.pl (T.Z.); p_ratajczak@ump.edu.pl (P.R.); kkus@ump.edu.pl (K.K.); 2Department of Internal Diseases, Metabolic Disorders and Arterial Hypertension, Poznan University of Medical Sciences, Szamarzewskiego 84 Street, 60-572 Poznań, Poland; karolinahoffmann@ump.edu.pl (K.H.); wieslawbryl@wp.pl (W.B.); 3Department of Computer Science and Statistics, Poznan University of Medical Sciences, Rokietnicka 7 Street, 60-806 Poznań, Poland; michal@ump.edu.pl; 4Department of Family Medicine, Wroclaw Medical University, 51-141 Wrocław, Poland; anna.hans.wytrychowska@gmail.com; 5Department of Pharmacology and Toxicology, Institute of Health Sciences, Collegium Medicum, University of Zielona Gora, Licealna 9 Street, 65-417 Zielona Góra, Poland; elapharm@ump.edu.pl

**Keywords:** pharmacovigilance, COVID-19, Pfizer–BioNTech (BNT162b2) vaccines, Moderna (mRNA-1273) vaccines, Oxford–AstraZeneca (ChAdOx1 nCoV-19) vaccines, SARS-CoV-2

## Abstract

The aim of the study was to compare the safety profiles (prevalence of both local and systemic side effects) of COVID-19 vaccines (Pfizer–BioNTech, Moderna, Oxford–AstraZeneca) among healthcare workers (doctors, nurses, and pharmacists) administered with a first and a second dose of the vaccines. Another goal of the research was to evaluate potential demographic and clinical risk factors for the frequency and intensity of side effects. A post-marketing, cross-sectional survey-based study was carried out on a sample of 971 respondents (323 doctors, 324 nurses, and 324 pharmacists), all more than 18 years old, who have taken two doses of the following SARS-CoV-2 vaccines: BNT162b2 (Pfizer–BioNTech) (group 1), mRNA-1273 (Moderna) (group 2), and ChAdOx1 nCoV-19 (Oxford–AstraZeneca) (group 3). A validated, self-administered questionnaire was developed and delivered online to the target population group of healthcare workers. The survey was conducted during the third wave of the COVID-19 (1 February 2021–1 July 2021) pandemic. It was based on the CAWI (computer-assisted web interview) method. Questionnaires were disseminated using selected social media. The BNT162b2 (Pfizer–BioNTech) vaccine was the most commonly administered COVID-19 vaccine among healthcare professionals in Poland (69.61%). Side effects following a SARS-CoV-2 vaccine were reported by 53.11% of respondents in group 1, 72% in group 2, and 67.59% in group 3. The following were the most common side effects regardless of the type of vaccine administered: pain at the injection site, headache, muscle pain, fever, chills, and fatigue. The number and intensity of reported side effects following administration of a BNT162b2 (Pfizer–BioNTech) vaccine were significantly lower than in the other two study groups (*p* < 0.00001). Risk factors for side effects following administration of one of the SARS-CoV-2 vaccines subject to the analysis included being female, young, and suffering from a diagnosed allergy. Our results clearly show that the short-term safety profiles of the eligible COVID-19 vaccines (Pfizer–BioNTech, Moderna, Oxford–AstraZeneca) are acceptable. Nevertheless, the two-dose COVID-19 vaccines available in Poland differ significantly in the frequency of both local and systemic side effects and their intensity. Women, young people, and patients diagnosed with allergies are particularly exposed to the risk of side effects. Further studies are needed to determine the long-term safety profile of COVID-19 vaccines.

## 1. Introduction

The first patient with coronavirus disease 2019 (COVID-19) was identified in China in December 2019 [1]. In December 2021, the cumulative number of confirmed COVID-19 cases across the world had reached nearly 266 million with the death toll exceeding 5 million [2]. The number of illnesses and fatalities is continuously increasing during the COVID-19 pandemic [3]. If adequate preventive actions are not performed quickly, COVID-19 will have serious and long-term medical, social, economic, and mental effects [4,5,6].

Effective vaccines are one of the most significant preventive measures to contain infectious diseases [7]. COVID-19 is the first disease where hundreds of institutions and companies are simultaneously engaged in research on producing effective vaccines from multiple platforms. By the end of 2021, more than 135 vaccines had entered clinical trials, with 13 in phase three clinical trials. Among those, mRNA vaccines (Pfizer–BioNTech and Moderna), recombinant adenovirus vectored vaccines (AstraZeneca, Cansino, Gamaleya, and Johnson Pharm), and inactivated vaccines (Sinopharm and Sinovac) have made the fastest progress [8]. At of the end of 2020, nine candidate vaccines had been authorized for human use in many countries [8,9]. On the last day of 2020, the WHO authorized Pfizer vaccines for emergency use worldwide, opening the door for mass COVID-19 vaccination programs [10].

Current evidence on the safety of COVID-19 vaccines relies mainly on data from phase one–three randomized controlled trials and vaccine safety surveillance systems in several countries [11]. In the opinion of many clinical researchers, additional post-authorization studies and long-term population-level surveillance are strongly encouraged to further define the safety profile of COVID-19 vaccines. An ideal vaccine should confer effective protection for a long time, possess a good safety profile, and should be affordable and easily accessible to all [12]. Based on the results of the published clinical reports for different COVID-19 vaccines, we have found that the currently developed COVID-19 vaccines differ significantly in their effectiveness and safety. Reports of suspected adverse drug reactions for these different therapies are an obvious top priority [3]. One of the first meta-analyses concerning the safety profile of COVID-19 vaccines was published by Wu et al. [13] in July 2021. They showed that the pooled rates of local and systemic reactions were significantly lower among inactivated vaccines (23.7%, 21.0%), protein subunit vaccines (33.0%, 22.3%), and DNA vaccines (39.5%, 29.3%), as compared with RNA vaccines (89.4%, 83.3%), non-replicating vector vaccines (55.9%, 66.3%), and virus-like particle vaccines (100.0%, 78.9%). Among local reactions, pain at the injection site was most frequently reported, while fatigue and headache were the most frequently reported systemic reactions. The authors noticed that the frequency of vaccine-related serious side effects was below 0.1% and balanced between the analyzed groups.

In Poland, the national SARS-CoV-2 vaccination program includes the use of four vaccines: BNT162b2 (Pfizer–BioNTech), mRNA-1273 (Moderna), ChAdOx1-S (Oxford–AstraZeneca), and Ad26.COV2.S (Johnson Pharm). The National Vaccination Program in Poland, like other government programs in the European Union (EU), has decided to prioritize healthcare workers in receiving the COVID-19 vaccine in the early stages of the national immunization strategy. Therefore, Polish healthcare decision makers have tried to ensure uninterrupted care and protect the most vulnerable patients. On 27 December 2020, the first cohort of health professionals received the COVID-19 vaccine. The implementation of the program began in Poland as a basic anti-COVID-19 strategy established by the Ministry of Health of the Republic of Poland [14].

The long-term effect of mass vaccination programs and related reports on their safety profile obtained from independent studies is also expected to positively impact many healthcare aspects indirectly affected by the COVID-19 pandemic, such as diagnosis and treatment of various conditions.

Accordingly, the presented study aimed to evaluate the short-term side effects (both local and systemic) experienced after a Pfizer–BioNTech (BNT162b2), a Moderna (mRNA-1273), or an Oxford–AstraZeneca (ChAdOx1 nCoV-19) vaccine among healthcare professionals (doctors, nurses, and pharmacists). Another goal of the research work was to identify selected clinical factors that have a significant impact on the risk of side effects following a SARS-CoV-2 virus vaccination.

## 2. Materials and Methods

### 2.1. Study Population

This post-marketing trial was designed as a cross-sectional survey-based study. The study group consisted of doctors, pharmacists, and nurses registered in the official database kept by the Supreme Chamber of Doctors and Nurses and Chief Pharmaceutical Inspectorate. Three thousand e-mails were sent to the above healthcare workers with a request to join the study voluntarily. The e-mail contained a link to the research questionnaire and all necessary information about the study’s purpose and rules of participation. Moreover, potential respondents were able to download the link to the study questionnaire from the Poznan University of Medical Sciences website promoting our study project. The recruitment process took place during February and March 2021 and was subsequently extended to July 2021. A total of 1130 respondents (all who agreed to participate in the study) were taken into consideration. However, based on the inclusion criteria of the study and incomplete questionnaires by 159 healthcare workers, 971 respondents were finally included in the study. The response rate, defined as the number of adequately completed online forms, was 86% (Figure 1).

The study group consisted of 971 respondents (323 doctors, 324 nurses, and 324 pharmacists), all more than 18 years old, who had taken two doses of the following SARS-CoV-2 vaccines: BNT162b2 (Pfizer–BioNTech), mRNA-1273 (Moderna), or ChAdOx1 nCoV-19 (Oxford–AstraZeneca), within the priority groups from January–April 2021 [14]. A research questionnaire with a request to fill it in electronically was sent to healthcare workers inviting them to take part in the study. Healthcare professionals meeting the following inclusion criteria were qualified for the study: 18 years old or more; in the process of receiving two doses of one of the following SARS-CoV-2 virus vaccines: BNT162b2 (Pfizer–BioNTech), mRNA-1273 (Moderna), or ChAdOx1 nCoV-19 (Oxford–AstraZeneca); time from the second dose of the vaccine not less than 4 weeks; and administration of two doses of one of the following SARS-CoV-2 virus vaccines: BNT162b2 (Pfizer–BioNTech), mRNA-1273 (Moderna), or ChAdOx1 nCoV-19 (Oxford–AstraZeneca), according to a vaccination schedule. If the inclusion criteria were not met, subjects were excluded from the study. The study began after the Bioethics Committee at the Poznan University of Medical Sciences issued an opinion (No. 25/22) confirming that the study had no features of a medical experiment. Before starting the survey, each participant was informed about the purpose of the project and that the study was safe, free, and anonymous, and consent to participate in the study could be withdrawn at any time, without giving any reason.

Patients were divided into three groups:**Group 1 (*n* = 676):** Respondents vaccinated with two doses of vaccine BNT162b2 (Pfizer–BioNTech) against SARS-CoV-2 virus;**Group 2 (*n* = 150):** Respondents vaccinated with two doses of vaccine mRNA-1273 (Moderna) against SARS-CoV-2 virus;**Group 3 (*n* = 145):** Respondents vaccinated with two doses of vaccine ChAdOx1 nCoV-19 (Oxford–AstraZeneca) against SARS-CoV-2 virus.

### 2.2. Study Technique

An anonymous questionnaire on Google Forms was constructed based on the scientific literature. It was validated and then implemented with the use of the CAVI (computer-assisted web interview) method. This method was chosen because of the pandemic and the applicable restrictions, to avoid any potential risks to the survey subjects. The Poznan University of Medical Sciences website and social media (e.g., Facebook, scientists association) were used to distribute questionnaires. Questionnaires were also distributed electronically (by e-mail) throughout the country. The questionnaire was developed following a standardized protocol that consisted of a literature review [15], focused group discussions, and an expert opinion. A pre-test procedure on a representative sample of 150 subjects (representative sample: 50 doctors, 50 nurses, and 50 pharmacists) was used as a suitable method for validation and for assessing the psychometric properties of the questionnaire. After the procedure, if necessary, the questions could be modified. However, the pre-test results were included in the post-test because the pre-test showed no need to modify the study tool. The internal consistency of the safety profile of COVID-19 vaccines questionnaire is based on the extent to which the items are correlated and was determined by calculating Cronbach’s α. Cronbach’s α was 0.79, which proves an excellent internal consistency. The study questionnaire was accepted by a group of 5 national medical consultants specializing in the field of infectious diseases.

The questionnaire comprised 25 questions divided into four categories:Demographic data (age, sex, height, weight, profession, and geographic region).Medical anamneses (chronic illnesses, health status, smoking, and alcohol consumption).COVID-19-related anamneses (type of vaccine, number of vaccine doses, dates of vaccine doses, previous infection, and diagnosis date).Vaccine side effects (local side effects, systemic side effects, onset, and duration).

The questionnaire inquired about the short-term side effects experienced following both doses of a COVID-19 vaccine. The side effects were classified as local or systemic, and their onset, duration, and intensity were self-assessed and self-reported by the participating subjects.

The primary outcome was a safety profile assessment (e.g., local and systemic reactions and side effects). Identification of selected clinical factors that have a significant impact on the risk of adverse effects following the administration of a SARS-CoV-2 vaccine was a secondary outcome.

All respondents who qualified for the study completed a safety profile assessment questionnaire for the selected SARS-CoV-2 vaccines twice. The first time was after the first dose of the vaccine and one day before the scheduled second dose of the vaccine. The second time was at least 28 days after receiving the 2nd dose of the SARS-CoV-2 virus vaccine.

### 2.3. Statistical Analysis

The quantitative parameters were presented using mean value, median, and standard deviation. Categorical data were presented as counts and percentages. A comparison of more than two groups was performed using the Kruskal–Wallis test with the post hoc Dunn’s test. The chi-square test for independence was used to analyze categorical data. In order to evaluate the potential demographic and medical predictors of side effects of administering COVID-19 vaccines, binary logistic regression for the incidence of local and systemic SRAEs was used. The inferential tests were done with the assumption for a confidence interval (CI) of 95% and a significance level of (*p*) < 0.05. Analysis was carried out using the TIBCO Software Inc. (2017) statistical package. Statistica (data analysis software system), version 13. http://statistica.io (accessed on 5 October 2021). All tests were considered significant at *p* < 0.05.

## 3. Results

### 3.1. Study Group Characteristics

A total of 1130 respondents were taken into consideration. However, based on the inclusion criteria of the study and incomplete filled in questionnaires by 159 healthcare workers, 971 respondents were finally included in the study. The study group consisted of 971 respondents (323 doctors, 324 nurses, and 324 pharmacists), all more than 18 years old, who had taken two doses of the following SARS-CoV-2 vaccines: BNT162b2 (Pfizer–BioNTech), mRNA-1273 (Moderna) or ChAdOx1 nCoV-19 (Oxford–AstraZeneca).

The characteristics of the study respondents are presented in Table 1. Individuals vaccinated with the BNT162b2 vaccine were most common amongst the respondents (*n* = 676, 69.61%), 150 respondents (15.48%) were vaccinated with the mRNA-1273 vaccine, while the ChAdOx1 nCoV-19 vaccine was administered to 145 respondents (14.93%). The analyzed groups of respondents did not differ in terms of gender, structure, age, health status, the frequency of presence of comorbidities, the frequency of stimulants use, or the frequency of infection with the SARS-CoV-2 virus (*p* > 0.05). There were significantly more subjects diagnosed with spine diseases in the group vaccinated with the ChAdOx1 nCoV-19 vaccine (11.03%, *p* = 0.04605) compared with the other two groups (group 1—5.47%; group 2—7.33%). Arterial hypertension was significantly more common in group 3 (13.10%, *p* = 0.04212), while hypothyroidism was substantially more frequent in group 2 (9.33%, *p* = 0.03243) (Table 1).

### 3.2. Safety Profile of COVID-19 Vaccines

As a result of the study, we found that the vast majority of respondents in each study group experienced side effects following the administration of one of the three SARS-CoV-2 vaccines subject to analysis. This percentage was highest in group 2 (72%) and lowest in group 1 (53.11%) (Table 2).

In the conducted study, no cases of anaphylaxis, thrombosis, and thrombocytopenia were observed after administering any of the analyzed COVID-19 vaccines (Table 2).

The most common local side effects in each study group was pain at the injection site. The above symptom was reported most frequently in the group following administration of the mRNA-1273 vaccine (72%) and least frequently by the group of respondents who had received the BNT162b2 vaccine (53.11%) (*p* < 0.00001). A total of 67.59% of subjects in group 3 reported pain at the injection site.

The most common systemic side effects following the BNT162b2 vaccine were fatigue (30.18%), headache (28.89%), muscle pain (25%), chills (19.67%), fever (16.57%), and joint pain (9.20%) (Table 2).

The most common systemic side effects following the mRNA-1273 vaccine were headache (50.00%), muscle pain (43.33%), fever (42.67%), fatigue (39.33%), chills (39.33%), joint pain (14.67%), lymphadenopathy (14.00%), and nausea (10.67%) (Table 2).

The most common systemic side effects following the ChAdOx1 nCoV-19 vaccine were fever (51.72%), muscle pain (46.90%), chills (46.21%), headache (42.76%), fatigue (33.10%), joint pain (24.14%), and nausea (11.72%) (Table 2).

Rash (skin-related SE prevalence) was reported only by 2.22% of respondents in group 1 and by 4.00% of respondents in group 2, *p* = 0.06056 (Table 2).

Pearson chi-square comparative analysis on the incidence of side effects between the three types of SARS-CoV-2 vaccines revealed statistically significant differences in the incidence of the following side effects: pain at the injection site (*p* = 0. 00009), swelling at the injection site (*p* < 0.00001), headache (*p* < 0.00001), muscle pain (*p* < 0.00001), chills (*p* < 0.00001), joint pain (*p* < 0.00001), fever (*p* < 0.00001), dizziness (*p* = 0.04945), nausea (*p* < 0.00001), lymphadenopathy (*p* < 0.00001), and diarrhea (*p* = 0.01620). Prevalence of both local and systemic side effects was significantly lower in group 1 (BNT162b2 vaccine) compared with the other two groups (mRNA-1273—group 2 and ChAdOx1 nCoV-19—group 3). Details of the comparative analysis of the incidence of adverse reactions following receipt of one of the three SARS-CoV-2 vaccines are shown in Table 2.

We used the Kruskal–Wallis test, to show that the SARS-CoV-2 vaccines analyzed were significantly different in terms of the number of side effects experienced following their administration (*p* < 0.00001). Group 1 respondents (BNT162b2 vaccine) reported the least number of side effects—median: 2.00 (0–5). The number of side effects in the other two groups was comparable: group 2—median: 5.00 (0–7); group 3—median: 5.00 (0–6) (*p* = 0.808387). The number of reported side effects following administration of a SARS-CoV-2 vaccine in group 1 was significantly lower than in the other two study groups (*p* < 0.00001).

On the basis of a chi-square test analysis, we found a correlation between the occurrence of side effects and the dose of a SARS-CoV-2 vaccine (*p* < 0.0001). Respondents in group 3 were significantly more likely to report side effects after receiving only one dose of the ChAdOx1 nCoV-19 vaccine (96.94%) compared with the other study groups (26.32%—group 1, 28.70%—group 2). In contrast, respondents in both group 1 and group 2 were significantly more likely to report side effects after both the first and second dose of a vaccine (42.94%—group 1, 46.30%—group 2) compared with group 3 respondents (3.06%) (Table 3).

Based on the research carried out, we found that side effects occurred on the first day after vaccination regardless of the type of vaccine in all the groups (group 1—84.17%, group 2—90.74%, group 3—86.73%, *p* = 0.52475) (Table 3).

In the Kruskal–Wallis test, there were no statistically significant differences between the type of SARS-CoV-2 vaccine and the mean duration of side effects (*p* = 0.6647). The mean duration of side effects to one of the three vaccines analyzed was: 2.53 ± 3.09—group 1, 2.31 ± 1.39—group 2, and 2.40 ± 2.84—group 3 (Table 3).

Across all study groups, respondents most commonly assessed the nature of the course of side effects as mild to moderate. Nevertheless, significant differences were observed between the type of SARS-CoV-2 vaccine administered and the nature of the severity of the experienced adverse reactions (*p* < 0.0001). Compared with the other two study groups (16.67%—group 2, 20.41%—group 3), group 1 respondents were significantly more likely to assess the severity of side effects as mild (37.40%). On the other hand, in both groups 2 and 3, respondents were significantly more likely to assess the nature of the course of side effects as moderate (71.30%—group 2, 66.33%—group 3) and severe (12.04%—group 2, 13.27%—group 3) when compared with group 1 (55.68%—moderate, 6.93%—severe) (Table 3).

In the case of the respondents vaccinated against COVID-19 based on mRNA technology (groups 1 and 2), a relationship between the duration of side effects and the confirmed SARS-CoV-2 infection at baseline was demonstrated. In both study groups, the duration of side effects after receiving the COVID-19 vaccine was significantly longer in people with confirmed SARS-CoV-2 virus infection at baseline than those who did not have a previously confirmed infection (4.8 ± 3.0 vs. 4.0 ± 3.1, *p* = 0.0364—group 1; 8.3 ± 2.5 vs. 5.7 ± 3.4, *p* = 0.0052—group 2). However, no correlation was observed between the confirmed SARS-CoV-2 infection at baseline and the number and intensity of side effects following the administration of SARS-CoV-2 virus vaccines among all study groups (*p* > 0.005) (Table 4).

### 3.3. Identification of Sociodemographic and Clinical Factors Influencing the Risk of Side Effects Following the Intake of SARS-CoV-2 Virus Vaccines

A logistic regression model of the effect of selected factors on the risk of side effects following administration of one of the analyzed SARS-CoV-2 vaccines showed that being female (*p* < 0.0001), the presence of an allergy (*p* = 0.005), and respiratory diseases (0.039) significantly influenced the risk of side effects following administration of the BNT162b2 vaccine. The risk of side effects after receiving the BNT162b2 vaccine was highest for women and those with a diagnosed allergy or respiratory disease (Table 5).

In the group of subjects vaccinated with the mRNA-1273 vaccine, being female (*p* = 0.040) and age (*p* = 0.007) were the two factors significantly affecting the risk of side effects. The risk of side effects after receiving the mRNA-1273 vaccine was highest for women and younger individuals (Table 5).

In the group of subjects vaccinated with the AZD1222 vaccine, the subject’s age (<0.0001), the presence of allergies (*p* = 0.017), and nicotine consumption (*p* = 0.008) significantly affected the risk of side effects. The risk of side effects after receiving the AZD1222 vaccine was highest for younger subjects and subjects with a diagnosed allergy and who did not consume nicotine (Table 5).

## 4. Discussion

A pandemic and unpredictable course of COVID-19, mortality rate above 7% in the 60–80 age group and close to 20% in the over 80 age group, difficult-to-understand and complex pathogenesis of the infection, lack of effective drugs, and the denial by certain communities of the existence of the SARS-CoV-2 virus are the basis and target for any activity required to stop the spread of infections [16]. Therefore, stopping the spread of the pandemic and preventing the associated deaths requires us to acquire population immunity quicker through mass vaccination. The primary goal is to achieve population immunity, which is guaranteed by patients who, for unknown, possibly genetic reasons, will never become infected, patients who have survived COVID-19, and patients who were subjected to vaccinations [17]. Vaccination is the safest way to acquire immunity to infection in a controlled manner. A vaccine is a biological substance that introduces antigens against which immunity is to be generated [13].

In Poland, the national SARS-CoV-2 vaccination program includes the use of four vaccines: BNT162b2 (Pfizer–BioNTech), mRNA-1273 (Moderna), ChAdOx1-S (Oxford–AstraZeneca), and Ad26.COV2.S (Johnson Pharm). A fifth vaccine will soon be available (NVX-CoV2373 by Novavax, an adjuvanted protein vaccine) for which the European Medicines Agency (EMA) has issued a positive scientific opinion for conditional marketing authorization on 20 December 2021 [14,18]. As of 4 February 2022, 58.31% of people in Poland have been vaccinated with the first dose of a coronavirus vaccine. A total of 49.9%of the total population is fully vaccinated [19].

During the COVID-19 pandemic, prompt research and simultaneous lack of follow-up time post-vaccination aroused great public concern about the safety profile of vaccine candidates. While rare and non-serious side effects should not derail mass vaccination, a thorough risk-benefit analysis should be done [20]. EMA’s detailed assessments take into account all available data from all sources to draw robust conclusions on the safety of the vaccines. These data include clinical trial results, reports of suspected side effects, epidemiological studies monitoring the safety of the vaccine, toxicological investigations, and any other relevant information [21].

According to the survey the vast majority of healthcare workers in Poland are vaccinated with Pfizer–BioNTech’s BNT162b2 (69.61%). This is due to the fact that Pfizer–BioNTech’s BNT162b2 vaccine was the first to be licensed in the EU, and the Polish government, as part of its strategy to implement a national COVID-19 vaccination program, classified healthcare workers as group 0, i.e., priority in the vaccination order. Furthermore, out of all vaccines available in the EU, the Polish government purchased the most doses of the Pfizer–BioNTech vaccine (42.9 million) [16]. The above figures are in line with an EMA report dated 20 January 2022, which shows that about 545 million doses of Comirnaty (Pfizer–BioNTech), 103 million doses of Spikevax (Moderna), and 69 million doses of Vaxzevria (Oxford–AstraZeneca) were administered in the EU/EEA between EU marketing authorization and 2 January 2022 [21]. These data are also consistent with a previous study conducted in Poland by Dziedzic et al. [22] on a group of 317 Polish healthcare professionals and students of the medical university where the safety profile of two types of SARS-CoV-2 vaccines: mRNA-based (Pfizer–BioNTech and Moderna) and viral vector-based (Oxford–AstraZeneca) were compared. In a study by Dziedzic et al. [22], similarly, 24.5% of participants received a viral vector-based vaccine and 77.5% of them received mRNA-based vaccines.

In this study, the occurrence of side effects following a SARS-CoV-2 vaccine administration was reported by 53.11% of respondents vaccinated with BNT162b2 (Pfizer–BioNTech), 72% of those vaccinated with mRNA-1273 (Moderna), and 67.59% of those vaccinated with ChAdOx1-S (Oxford–AstraZeneca). The most common local or systemic side effects regardless of the type of vaccine received were pain at the injection site (49.93%—BNT162b2, 69.33%—mRNA-1273, 53.10%—ChAdOx1-S), headache (28.89%—BNT162b2, 50.00%—mRNA-1273, 42.76%—ChAdOx1-S), muscle pain (25.00%—BNT162b2, 43.33%—mRNA-1273, 46.90%—ChAdOx1-S), fever (16.57%—BNT162b2, 42.67%—mRNA-1273, 51.72%—ChAdOx1-S), chills (19.67%—BNT162b2, 39.33%—mRNA-1273, 46.21%—ChAdOx1-S) and fatigue (30.18%—BNT162b2, 39.33%—mRNA-1273, 33.10%—ChAdOx1-S). These observations are in line with the results reported in phase three clinical trials and vaccine fact sheets [23,24,25,26,27,28,29].

In clinical trials evaluating the efficacy and safety profile of the BNT162b2 vaccine (Pfizer–BioNTech), local and systemic reactions occurring within 7 days of an injection were reported by 66% and 50% of subjects, respectively. The most common side effects in participants aged 16 years or older were pain at the injection site (>80%), fatigue (>60%), headache (>50%), muscle pain and chills (>30%), joint pain (>20%), and fever and swelling at the injection site (>10%) [23]. More BNT162b2 recipients than placebo recipients reported any side effects (27% and 12%, respectively) or related side effects (21% and 5%). This distribution primarily reflects transient reactogenicity events that were reported as side effects more commonly by vaccine recipients than placebo recipients. However, the incidence of serious adverse events was low and was similar in the vaccine and placebo groups [23]. In clinical trials evaluating the efficacy and safety profile of the mRNA-1273 vaccine (Moderna), local and systemic reactions were reported by 86% and 67% of subjects, respectively. The most common side effects experienced by participants aged 18 years or older after both the first and the second dose of the vaccine were pain at the injection site (>87%), fatigue (>67%), and headache (>60%) [24]. Solicited side effects at the injection site occurred more frequently in the mRNA-1273 group than in the placebo group after both the first dose (84.2% vs. 19.8%) and the second dose (88.6% vs. 18.8%) Similarly, solicited systemic side effects occurred more often in the mRNA-1273 group than in the placebo group after both the first dose (54.9% vs. 42.2%) and the second dose (79.4% vs. 36.5%). Serious adverse events were rare, and the incidence was similar in the vaccine and placebo groups [24].

In clinical trials evaluating the efficacy and safety profile of the ChAdOx1 nCoV-19 vaccine (Oxford–AstraZeneca), 60% of subjects reported the occurrence of local and systemic side effects. The most commonly reported side effects were tenderness at the injection site (63.7%), pain at the injection site (54.2%), headache (52.6%), fatigue (53.1%), muscle pain (44.0%), malaise (44.2%), fever (33.6%), chills (31.9%), joint pain (26.4%), and nausea (21.9%) [25]. A total of 175 severe adverse events occurred in 168 participants, 84 events in the ChAdOx1 nCoV-19 group and 91 in the control group. Three events were classified as possibly related to a vaccine: one in the ChAdOx1 nCoV-19 group, one in the control group, and one in a participant who remains masked to group allocation [25].

In addition, the results obtained are consistent with previous observational studies conducted in Poland and worldwide. In a Polish study mentioned above, 78.9% and 60.7% of the individuals reported at least one local and one systemic side effect after the first and second dose of the COVID-19 vaccine, respectively [22]. There were observed various side effects, including pain at the injection site (76.9%), and systemic-like fatigue (46.2%), headache (37.7%), and muscle pain (31.6%). It must be pointed out that 35.2% of local and 44.8% of systemic side effects subsided up to 1 day after inoculation with both types of vaccines. Dziedzic et al. [22] evaluated that mRNA-based vaccines caused a higher prevalence of local side effects, mainly pain at the injection site (81.3% vs. 71.7%; *p* = 0.435), whereas viral vector-based vaccines caused mainly mild systemic side effects (76.7% vs. 55.3%; *p* = 0.004) after both doses. Research conducted in the U.K. and Saudi Arabia also confirms the obtained results [26,27]. Both studies aimed to evaluate the short-term side effects experienced after receiving either a Pfizer–BioNTech mRNA (BNT162b2) or an Oxford–AstraZeneca (ChAdOx1 nCoV-19) vaccine in general populations. Both this study and the studies by Alhazmi et al. [26] and Menni et al. [27] showed that the vast majority of subjects reported side effects regardless of the type of vaccine received. The most common local side effects included pain at the injection site. Systemic side effects included headache and fatigue. Research carried out thus far around the world as well as this report has shown that participants who were vaccinated with an Oxford–AstraZeneca vaccine are at a greater risk of systemic side effects, including fatigue and fever, compared with those who received a Pfizer–BioNTech vaccine [22,26,27,30].

This study found that the majority of local and systemic side effects, regardless of the type of vaccine received, were mild to moderate in severity and were usually resolved within 2–3 days of their onset. The vast majority of side effects, irrespective of the type of vaccine, occurred on the first day after vaccination. For Pfizer and Moderna vaccines, the highest proportion of subjects reported experiencing side effects after both the first and second dose: 42.94% and 46.30%, respectively. For the Oxford–AstraZeneca vaccine, however, side effects after the first dose were reported significantly more frequently than after the second dose (96.94% vs. 0.00%). It was proved, both in clinical trials and in the real-world data, that side effects associated with the COVID-19 vaccine are mild to moderate [23,24,25,26,27,28,29]. In a phase three clinical trial for Pfizer’s vaccine, it was shown that in general, local and systemic reactions were mostly mild-to-moderate in severity and were observed within the first 1 to 2 days after vaccination and resolved within 1 to 2 days. The proportion of participants reporting local and systemic reactions did not increase after the second dose [23]. Moderna’s vaccine phase three clinical study showed that injection-site events were mainly grade 1 or 2 in severity and lasted a mean of 2.6 and 3.2 days after the first and second doses, respectively. The severity of the solicited systemic events increased after the second dose in the mRNA-1273 group, with an increase in the proportions of grade 2 events (from 16.5% after the first dose to 38.1% after the second dose) and grade 3 events (from 2.9% to 15.8%). Solicited systemic side effects in the mRNA-1273 group lasted a mean of 2.9 days and 3.1 days after the first and second doses, respectively [24]. A phase three clinical trial of the Oxford–AstraZeneca vaccine showed that it is tolerated and that the side effects are less both in intensity and number in older adults, with lower doses, and after the second dose [25].

The study found that being female, young, and suffering from a diagnosed allergy are all risk factors for side effects following administration of one of the SARS-CoV-2 vaccines subject to analysis. The obtained results are consistent with both clinical and observational studies [22,23,24,25,26,27]. For example, Menni et al. [27] recruited participants at the mean age of 50 y, and most of them were above 55 y. Those individuals reported tiredness less frequently (between 8% and 21% of the study group) and female subjects experienced more side effects than male ones. Similarly, Dziedzic et al. [22] pointed out that the most important factors predisposing to side effects after both an mRNA-based vaccine or a viral vector-based vaccine were being female, young, presence of comorbidities, and systematic use of medications (especially palliative drugs consumption). Moderna vaccine’s phase three clinical trials further demonstrated that solicited adverse events were less common in participants who tested positive for a SARS- CoV-2 infection at baseline than in those who were negative at baseline [24]. Our study observed a relationship between the duration of side effects and the confirmed SARS-CoV-2 infection at baseline only among respondents vaccinated against COVID-19 based on mRNA technology (Pfizer, Moderna). In both study groups, the duration of side effects after receiving the COVID-19 vaccine was significantly longer in people with confirmed SARS-CoV-2 virus infection at baseline than those who did not have a previously confirmed infection.

Similar to other medications, allergic reactions can occur during vaccination. While most reactions are neither frequent nor serious, anaphylactic reactions are potentially life-threatening allergic reactions that are encountered rarely but can cause serious complications. Reactions are more often caused by inert substances, called excipients, which are added to vaccines to improve stability and absorption, increase solubility, influence palatability, or create a distinctive appearance, and not by the active vaccine itself. The excipients mostly incriminated for allergic reactions are polyethylene glycol, also known as macrogol, found in the currently available Pfizer–BioNTech and Moderna COVID-19 mRNA vaccines, and polysorbate 80, also known as Tween 80, present in Oxford–AstraZeneca and Johnson & Johnson COVID-19 vaccines [31]. Therefore, people suffering from allergies have a greater risk of side effects following a COVID-19 vaccination.

Differences in the safety profiles of vaccines should always be discussed in the context of their efficacy. Currently, efficacy data confirm that all available vaccines exceed the 50% threshold set by WHO [17] and can significantly reduce the number of symptomatic cases, hospitalizations, severe diseases, and death [32,33]. The effectiveness of mRNA-based vaccines in reducing infection incidence of COVID-19 after the second dose, irrespective of prior infection with SARS-CoV-2, is comparable (95%—Pfizer–BioNTech; 94.1%—Moderna). In turn, the clinical effectiveness of the analyzed adenovirus-vectored vaccine (Oxford–AstraZeneca) in the prevention of COVID-19 is 70.4% among subjects receiving the two recommended doses at any dosing intervals (between 3 and 23 weeks) [23,24,25,34]. Moreover, many performed studies have shown that a number of factors can influence the observed difference in efficacy and effectiveness of the COVID-19 vaccines. Vaccine effectiveness can be affected by the following factors: population host factors (e.g., those who were not included in clinical trials) and virus factors (e.g., variants), as well as programmatic factors (e.g., adherence to dosing schedules or vaccine storage/handling) [35].

As of 28 December 2021, a total of 8,687,201,202 vaccine doses have been administered [2]. This mass vaccination should facilitate the identification of more uncommon and rare adverse events following immunization (AEFI). On the basis of data from the Vaccine Adverse Event Reporting System (VAERS) and the V-safe system of the U.S. Centers for Disease Control and Prevention (CDC), the rates of non-serious AEFI after public administration of BNT162b2 and mRNA-1273 were similar to the clinical trials [36]. Anaphylaxis occurs at a rate of approximately 1 case per million doses for the majority of vaccines, whereas the rates of anaphylaxis associated with BNT162b2 and mRNA1273 appear to be 4.7 times and 2.5 times higher, respectively [37,38]. Thrombosis and thrombocytopenia as a side effect of adenoviral vector vaccines (ChAdOx1 nCoV-19 and Ad26.COV2.S) were noted, including several deaths and severe outcomes [39,40,41,42]. Neither anaphylaxis following a BNT162b2 or a mRNA1273 vaccine, nor thrombosis and thrombocytopenia following a ChAdOx1 nCoV-19 vaccine were observed in this study.

There is some evidence that two COVID-19 mRNA vaccines (BNT162b2 and mRNA-1273) are safe in certain populations such as pregnant women [43] immunosuppressive patients [44], and HIV-positive subjects [45].

Further studies are needed to determine the long-term safety profile of COVID-19 vaccines. Further studies will also facilitate improved vaccine recommendations, vaccine safety surveillance systems, monitoring of early COVID-19 vaccine recipients, standardized reporting, and pharmacovigilance mechanisms. Safety issues noted for mass vaccination may have a deleterious impact on the global vaccine supply and the already fragile confidence in vaccines. Government agencies and scientists working on the development of COVID-19 vaccines should widely present any safety issues noted in mass vaccinations in order to reduce public vaccine hesitancy. Mass COVID-19 vaccination seems to be one of the most important ways to allow a worldwide return to normal life.

Despite it being pioneering research in Poland to discuss the side effects related to COVID-19 vaccines, there are some limitations. The primary weakness of this study is related to self-reported data and information about COVID-19 vaccine side effects, instead of objective information reported by healthcare workers following formalized, unbiased assessments. The authors expected that healthcare workers were more likely to be cautious about health issues, as compared with the general population, as this group, by default, has adequate levels of health literacy, as well as scientific interest. Another limitation was the predominance of respondents vaccinated by a BNT162b2 (Pfizer–BioNTech) COVID-19 vaccine, which resulted from phase one administration of this vaccine among healthcare professionals in the first weeks of 2021. Such a procedure was ordered by decision makers and public health officials. We decided to conduct this research project as a web-based study to ensure maximum safety for all study participants during the COVID-19 pandemic. Therefore, data were collected online in the form of a self-assessment survey. This might have compromised the inclusion of healthcare workers with limited internet access. However, the presented study is a good source of information on the occurrence of short-term adverse events of the most frequently administered COVID-19 vaccines. Therefore, the presented study meets the information needs of healthcare policymakers and the whole medical community.

## 5. Conclusions

Our results clearly show that the short-term safety profiles of the eligible COVID-19 vaccines (Pfizer–BioNTech, Moderna, Oxford–AstraZeneca) are acceptable. The incidence, duration, and nature of the severity of side effects reported by the recruited subjects are similar to those reported in clinical trials, indicating all three vaccines have safe profiles. Nevertheless, the two-dose COVID-19 vaccines available in Poland differ significantly in the frequency of both local and somatic side effects and their intensity. Women, young people, and patients diagnosed with allergies are particularly exposed to the risk of side effects. Further studies are needed to determine the long-term safety profile of COVID-19 vaccines.

## Figures and Tables

**Figure 1 vaccines-10-00434-f001:**
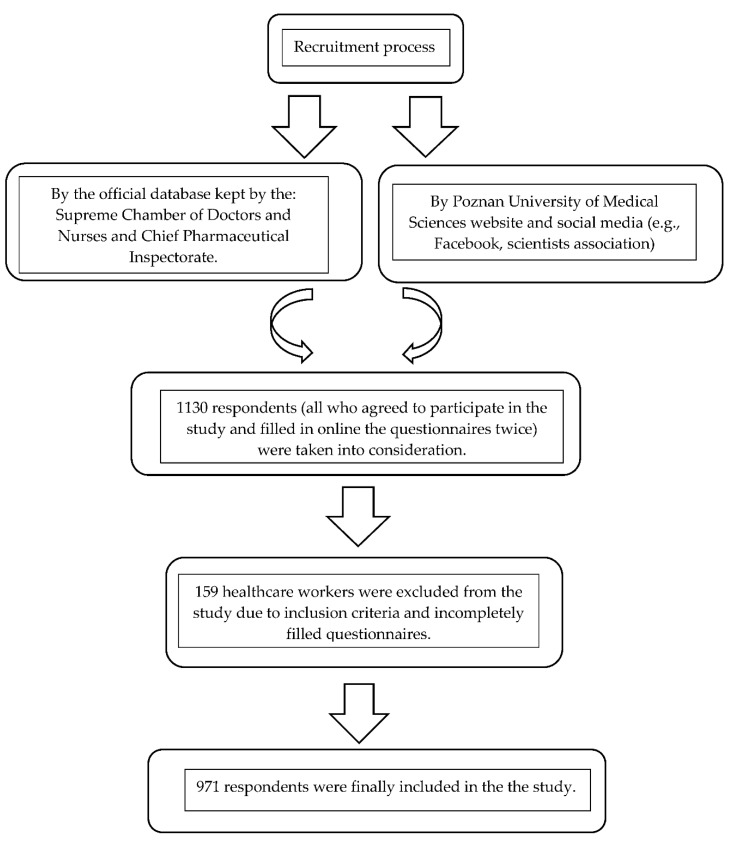
Recruitment process and applied research methods.

**Table 1 vaccines-10-00434-t001:** Comparative characteristics of the surveyed respondents vaccinated against the SARS-CoV-2 virus (*n* = 971).

Variable	Type of SARS-CoV-2 Virus Vaccine			*p* Value
	BNT162b2 (Group 1)	mRNA-1273 (Group 2)	ChAdOx1 nCoV-19 (Group 3)	
Group size *n* (%)	676 (69.62%)	150 (15.45%)	145 (14.93%)	
Gender (%):				0.99153 *
Women	78.25	78.00	78.62	
Men	21.75	22.00	21.38	
Age (Mean ± SD)	33.41 ± 14.77	34.74 ± 14.03	36.32 ± 15.08	0.06781 ^#^
Self-estimated health status (%):				0.18675 *
Excellent	11.83	10.67	6.21	
Very good	50.44	49.33	48.28	
Good	34.02	34.00	38.62	
Not so good	3.25	4.67	6.90	
Poor	0.44	1.33	0.00	
Presence of comorbidities (%)	46.01	39.33	46.21	0.31541 *
Most common comorbidity types (%):				
Spine diseases	5.47 ^a^	7.33 ^a^	11.03 ^b^	0.04605 *
Heart failure	1.92	2.00	2.76	0.81208 *
Obesity	5.18	9.33	4.83	0.12380 *
Hashimoto’s disease	3.25	3.33	1.38	0.46709 *
Hypothyroidism	7.25 ^a^	9.33 ^a^	2.07 ^b^	0.03243 *
Depression	2.07	4.00	2.76	0.37660 *
Allergy	16.27	13.33	15.86	0.67027 *
Arterial Hypertension	9.76 ^a^	4.67 ^b^	13.10 ^a^	0.04212 *
Diabetes	4.44	3.33	2.76	0.58041 *
Respiratory system diseases	3.11	1.33	3.45	0.45559 *
Previous infection with SARS-CoV-2 virus (%)	10.06	7.33	15.17	0.07648 *
Use of stimulants (%):				
Alcohol	41.42	44.00	33.10	0.11724 *
Nicotine	13.31	14.00	12.41	0.92193 *

*—The chi-square for independence; a,b—groups followed by the same letter do not differ statistically significantly; ^#^—the Kruskal–Wallis test.

**Table 2 vaccines-10-00434-t002:** Assessment of the frequency of side effects of COVID-19 vaccines reported by Polish healthcare workers. (*n* = 971).

Type of Side Effects (%)		Type of SARS-CoV-2 Virus Vaccine			*p* Value
		BNT162b2 (Group 1)	mRNA-1273 (Group 2)	ChAdOx1 nCoV-19 (Group 3)	
General		53.11 ^a^	72.00 ^b^	67.59 ^b^	<0.00001 *
Local SE Prevalence	Pain at the injection site	49.93 ^a^	69.33 ^b^	53.10 ^a^	0.00009 *
	Swelling at the injection site	8.47 ^a^	19.33 ^b^	3.45 ^a^	<0.00001 *
Systemic SE Prevalence	Headache	28.89 ^a^	50.00 ^b^	42.76 ^b^	<0.00001 *
	Muscle pain	25.00 ^a^	43.33 ^b^	46.90 ^b^	<0.00001 *
	Chills	19.67 ^a^	39.33 ^b^	46.21 ^b^	<0.00001 *
	Fatigue	30.18	39.33	33.10	0.08993 *
	Shortness of Breath	0.15	0.67	0.00	0.37552 *
	Problems with concentration	0.30	1.33	0.69	0.26254 *
	Joint pain	9.20 ^a^	14.67 ^b^	24.14 ^c^	<0.00001 *
	Fever	16.57 ^a^	42.67 ^b^	51.72 ^b^	<0.00001 *
	Dizziness	0.30 ^a^	2.00 ^b^	1.38 ^b^	0.04945 *
	Nausea	2.51 ^a^	10.67 ^b^	11.72 ^b^	<0.00001 *
	Lymphadenopathy	4.88 ^a^	14.00 ^b^	2.07 ^a^	<0.00001 *
	Diarrhea	0.30 ^a^	0.00 ^a^	2.07 ^b^	0.01620 *
	Insomnia	0.44	0.67	1.38	0.42562 *
	Sweating	0.15	0.00	0.69	0.35510 *
	Anaphylaxis	0.00	0.00	0.00	N/A
	Thrombosis	0.00	0.00	0.00	N/A
	Thrombocytopenia	0.00	0.00	0.00	N/A
Skin-related SE Prevalence	Rash	2.22	4.00	0.00	0.06056 *

*—The chi-square for independence; a,b,c—groups followed by the same letter do not differ statistically significantly; SE—side effects.

**Table 3 vaccines-10-00434-t003:** Duration and nature of the intensity of side effects following the administration of SARS-CoV-2 virus vaccines among healthcare professionals in Poland (*n* = 971).

Variable	Type of SARS-CoV-2 Virus Vaccine			*p* Value
	BNT162b2 (Group 1)	mRNA-1273 (Group 2)	ChAdOx1 nCoV-19 (Group 3)	
Vaccine dose followed by side effects (%):				<0.0001 *
1 dose	26.32 ^a^	28.70 ^a^	96.94 ^b^	
2 dose	30.75 ^a^	25.00 ^a^	0.00 ^b^	
1 dose and 2 dose	42.94 ^a^	46.30 ^a^	3.06 ^b^	
Time the side effects started to appear (%):				0.52475 *
1 day after vaccination	84.17	90.74	86.73	
2 days after vaccination	12.78	8.33	13.27	
3 days after vaccination	1.39	0.00	0.00	
4 days after vaccination	1.11	0.00	0.00	
5 days after vaccination	0.28	0.00	0.00	
6 days after vaccination	0.28	0.93	0.00	
Duration of side effects (days) (Mean ± SD ^&^)	2.53 ± 3.09	2.31 ± 1.39	2.40 ± 2.84	0.6647 ^#^
Nature of the intensity of side effects (%):				<0.0001 *
mild	37.40 ^a^	16.67 ^b^	20.41 ^b^	
moderate	55.68 ^a^	71.30 ^b^	66.33 ^b^	
severe	6.93 ^a^	12.04 ^b^	13.27 ^b^	

*—The chi-square for independence; a,b—groups followed by the same letter do not differ statistically significantly; ^#^—the Kruskal–Wallis test; ^&^—standard deviation.

**Table 4 vaccines-10-00434-t004:** Influence of the confirmed SARS-CoV-2 infection at baseline on number and intensity of side effects following the administration of SARS-CoV-2 virus vaccines among healthcare professionals in Poland (*n* = 971).

Variable	Type of SARS-CoV-2 Virus Vaccine								
	BNT162b2 (Group 1)			mRNA-1273 (Group 2)			ChAdOx1 nCoV-19 (Group 3)		
	Confirmed SARS-CoV-2 Infection at Baseline			Confirmed SARS-CoV-2 Infection at Baseline			Confirmed SARS-CoV-2 Infection at Baseline		
	Yes	No	*p* Value	Yes	No	*p* Value	Yes	No	*p* Value
Number of side effects (Mean ± SD)	6.3 ± 3.2	5.6 ± 3.8	0.2823 *	5.9 ± 4.2	6.3 ± 3.8	0.7269 *	5.6 ± 3.3	6.2 ± 3.5	0.5807 *
Duration of side effects (days) (Mean ± SD)	4.8 ± 3.0	4.0 ± 3.1	0.0364 *	8.3 ± 2.5	5.7 ± 3.4	0.0052 *	4.6 ± 3.4	5.7 ± 3.4	0.1994 *
Nature of the intensity of side effects *n* (%):									
mild	18 (40.0)	117 (37.0)		1 (9.1)	17 (17.5)	0.278 ^#^	4 (30.8)	16 (18.8)	0.551 ^#^
moderate	23 (51.1)	178 (56.3)	0.751 ^#^	10 (90.9)	67 (69.1)		8 (61.5)	57 (67.1)	
severe	4 (8.9)	21 (6.7)		0 (0.0)	13 (13.4)		1 (7.7)	12 (14.1)	

*—The U Mann–Whitney test; ^#^—the chi-square for independence; SD—standard deviation.

**Table 5 vaccines-10-00434-t005:** Logistic regression model for the risk of side effects following administration of SARS-CoV-2 virus vaccines among healthcare workers in Poland (*n* = 971).

	Risk of Side Effects					
	BNT162b2 (Group 1)		mRNA-1273 (Group 2)		ChAdOx1-S (Group 3)	
Variable	Coeff. (95% CI)	*p* Value	Coeff. (95% CI)	*p* Value	Coeff. (95% CI)	*p* Value
Gender:						
Female ref.						
Male	0.47 (0.32, 0.69)	<0.0001	0.42 (0.18, 0.96)	0.040	0.83 (0.36, 1.93)	0.681
Age	0.99 (0.97, 1.00)	0.056	0.96 (0.94, 0.99)	0.007	0.93 (0.90, 0.96)	<0.0001
Self-estimated health status:						
Excellent ref.						
Very good	0.62 (0.37, 1.01)	0.058	2.41 (0.50, 11.63)	0.271	0.74 (0.16, 3.28)	0.697
Good	1.19 (0.85, 1.66)	0.310	0.75 (0.34, 1.66)	0.486	0.72 (0.33, 1.55)	0.410
Not so good	1.27 (0.53, 3.06)	0.585	0.46 (0.09, 2.24)	0.338	0.24 (0.06, 0.55)	0.086
Poor	1.13 (0.91, 1.39)	0.256	2.89 (1.21, 2.41)	0.516	1.22 (0.89, 1.66)	0.309
Most common comorbidity types:						
Hashimoto’s disease	2.41 (0.93, 6.25)	0.069	1.57 (0.17, 14.53)	0.688	0.47 (0.02, 7.75)	0.601
Allergy	1.84 (1.20, 2.82)	0.005	1.65 (0.51, 5.26)	0.396	6.13 (1.37, 27.39)	0.017
Obesity	1.05 (0.53, 2.08)	0.886	1.47, 0.39, 5.57)	0.567	2.99 (0.35, 25.66)	25.66
Heart failure	1.03 (0.34, 3.09)	0.957	0.77 (0.06, 8.76)	0.836	0.46 (0.06, 3.43)	0.456
Diabetes	0.87 (0.42, 1.82)	0.727	1.57 (0.17, 14.53)	0.688	0.15 (0.01, 1.49)	0.106
Hypothyroidism	1.56 (0.86, 2.86)	0.142	1.47 (0.39, 5.57)	0.567	0.95 (0.08, 10.84)	0.973
Respiratory system diseases	2.91 (1.05, 8.03)	0.039	0.88 (0.51, 1.96)	0.866	1.95 (0.21, 18.01)	0.553
Spine diseases	1.06 (0.73, 1.89)	0.856	1.57 (0.87, 2.66)	0.155	1.86 (0.99, 2.01)	0.884
Use of stimulants:						
Alcohol	1.05 (0.77, 1.43)	0.719	1.06 (0.51, 2.19)	0.860	1.08 (0.51, 2.27)	0.833
Nicotine	0.70 (0.45, 1.10)	0.124	0.96 (0.34, 2.68)	0.950	0.25 (0.09, 0.70)	0.008

## Data Availability

The datasets used and/or analyzed during the current study are available from the corresponding author on reasonable request.

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
