# Peer review of "Safety Profile of COVID-19 Vaccines among Healthcare Workers in Poland"

_vaccines, 2022, doi:10.3390/vaccines10030434_

Round 1

Reviewer 1 Report

The current manuscript by Paczkowska, et al, entitled “Pharmacovigilance of COVID-19 vaccines among healthcare workers in Poland” examines the safety COVID-19 vaccines (Pfizer/BionTech, Moderna, Oxford-Astra Zeneca) among healthcare workers (doctors, nurses and pharmacists) in Poland. The study enrolled 971 respondents and used an online survey administered during the third wave of the pandemic in Poland. The authors report “adverse events”53.11%, 72% and 67.59% for Pfizer, Moderna and Astra-Zeneca respectively. The manuscript is of interest because is presents comparative data on side effects of the three most commonly used COVID-19 vaccines worldwide.  However, even the title is a bit misleading and does not convey what the data actually shows.  The authors conclude that the current vaccines are safe and effective, this could be the title of the manuscript.

General and Specific areas of concern:

  1. The authors tone in this manuscript is a bit troubling because they insist on using “adverse reactions” to describe common, mostly mild side effects that anyone who gets a vaccine exhibits. This is a small study of a very select demographic in a single country. The authors use side effects and adverse events interchangeably, which is technically not incorrect, but in the current world of anti-vaccine sentiments it is important to be more thoughtful on how one describes common side effects and not give the impression that these vaccines are causing significant unexpected adverse reactions.

  1. On lines 41-43 the authors state “However, the incidence of specific adverse reactions, their severity and the dose sequence after which they occurred were significantly different for the various analysed SARS-CoV-2 vac cines (p<0.005).” This statement in the abstract is without context and should be modified to reflect that Pfizer had lower numbers of specific side effects compared to the other tow vaccines. It should also be acknowledged here that the Pfizer vaccine was the most frequently administered in this population.

  1. Lines 90-93: The authors infer that safety needs to be addressed and that epidemiological tools are lacking. He authors have generalized these comments. However, these issues are not global. Epidemiological tools are widely available and are being used worldwide. Instigating the safety of the vaccines is an ongoing process that has been in place for other vaccines and is being implemented during the current vaccine roll out.

  1. I recommend changing the majority of references o “adverse reactions” to “side effects” for balance.

  1. From lines 133 to 142, the authors write about anaphylaxis and report that they can be rear events. However, they did not have any patients in this cohort with allergic reactions or anaphylaxis. Why not start with that here before this paragraph?

  1. Lines 145 to 146 discuss efficacy of the covid-19 vaccines and the authors state that thy are above the 50% efficacy. He authors spent a lot of time writing about other things that were not essential and wrote several paragraphs about stats that are irrelevant to the data they have presented here. Maybe you should say what the individual efficacies of these current vaccines are.

  1. On line 157 to 158 the authors assert that adenoviral vector vaccines lead to serious side effects “including several deaths and severe outcomes (39,40). However, these references talk about thrombocytopenia and no deaths. I did not see any references for several deaths. That has also not been reported anywhere else. The VAERS data base says there were 9 deaths associated with J&J. What is your number here? Where are your references?

  1. The study also did not look at the frequency of side effects in individuals who got placebos in clinical trials. This comparison would have added a valuable perspective for the readers since it would better gauge what was caused by the vaccinate versus the injection.

Author Response

Reviewer 1.

We would like to thank for a great work done during thoroughly reviewing of our article.

Comments:

The current manuscript by Paczkowska, et al, entitled “Pharmacovigilance of COVID-19 vaccines among healthcare workers in Poland” examines the safety COVID-19 vaccines (Pfizer/BionTech, Moderna, Oxford-Astra Zeneca) among healthcare workers (doctors, nurses and pharmacists) in Poland. The study enrolled 971 respondents and used an online survey administered during the third wave of the pandemic in Poland. The authors report “adverse events”53.11%, 72% and 67.59% for Pfizer, Moderna and Astra-Zeneca respectively. The manuscript is of interest because is presents comparative data on side effects of the three most commonly used COVID-19 vaccines worldwide.  However, even the title is a bit misleading and does not convey what the data actually shows.  The authors conclude that the current vaccines are safe and effective, this could be the title of the manuscript.

Response: Thank You for your valuable comment. We have changed the title of the paper as follows: Safety profile of COVID-19 vaccines among healthcare workers in Poland.  

General and Specific areas of concern:

  1. The authors tone in this manuscript is a bit troubling because they insist on using “adverse reactions” to describe common, mostly mild side effects that anyone who gets a vaccine exhibits. This is a small study of a very select demographic in a single country. The authors use side effects and adverse events interchangeably, which is technically not incorrect, but in the current world of anti-vaccine sentiments it is important to be more thoughtful on how one describes common side effects and not give the impression that these vaccines are causing significant unexpected adverse reactions.

Response: Thank You for your valuable comment. We have changed the majority of references of “adverse reactions” to “side effects” for balance.

2. On lines 41-43 the authors state “However, the incidence of specific adverse reactions, their severity and the dose sequence after which they occurred were significantly different for the various analysed SARS-CoV-2 vac cines (p<0.005).” This statement in the abstract is without context and should be modified to reflect that Pfizer had lower numbers of specific side effects compared to the other tow vaccines. It should also be acknowledged here that the Pfizer vaccine was the most frequently administered in this population.

Response: Thank You for your valuable comment. We have changed the above statement in the abstract as follows: “The BNT162b2 (Pfizer-BioNTech) vaccine was the most commonly administered COVID-19 vaccine among healthcare professionals in Poland (69.61%). The number and intensity of reported side effects following administration of a BNT162b2 (Pfizer-BioNTech) vaccine was significantly lower than in the other two study groups (p<0.00001).”

3. Lines 90-93: The authors infer that safety needs to be addressed and that epidemiological tools are lacking. He authors have generalized these comments. However, these issues are not global. Epidemiological tools are widely available and are being used worldwide. Instigating the safety of the vaccines is an ongoing process that has been in place for other vaccines and is being implemented during the current vaccine roll out.

Response: Thank You for your valuable comment. We have deleted the above statement in lines 90-93. In discussion section we are writing that “EMA’s detailed assessments take into account all available data from all sources to draw robust conclusions on the safety of the vaccines. These data include clinical trial results, reports of suspected side effects, epidemiological studies monitoring the safety of the vaccine, toxicological investigations and any other relevant information”.

4. I recommend changing the majority of references o “adverse reactions” to “side effects” for balance.

Response: Thank You for your valuable comment. We have changed the majority of references of “adverse reactions” to “side effects” for balance.

5. From lines 133 to 142, the authors write about anaphylaxis and report that they can be rear events. However, they did not have any patients in this cohort with allergic reactions or anaphylaxis. Why not start with that here before this paragraph?

Response: Thank You for your valuable comment. We have changed the above statement as follows: “In the conducted study, no cases of anaphylaxis, thrombosis, and thrombocytopenia were observed after administering any of the analyzed COVID-19 vaccines”. Moreover, as suggested by the reviewer, we included these data at the beginning of the discussion of the safety profile of the analyzed Covid-19 vaccines.

6. Lines 145 to 146 discuss efficacy of the covid-19 vaccines and the authors state that thy are above the 50% efficacy. He authors spent a lot of time writing about other things that were not essential and wrote several paragraphs about stats that are irrelevant to the data they have presented here. Maybe you should say what the individual efficacies of these current vaccines are.

Response: Thank You for your valuable comment. As suggested by the reviewer, the above part of the discussion was enriched with the clinical effectiveness of the analyzed vaccines and the analysis of factors influencing the differences in the effectiveness of the available vaccines against Covid-19.

7. On line 157 to 158 the authors assert that adenoviral vector vaccines lead to serious side effects “including several deaths and severe outcomes (39,40). However, these references talk about thrombocytopenia and no deaths. I did not see any references for several deaths. That has also not been reported anywhere else. The VAERS data base says there were 9 deaths associated with J&J. What is your number here? Where are your references?

Response: Thank You for your valuable comment. As suggested by the reviewer, we have completed the scientific references regarding the occurrence of fatal cases of patients who presented with venous thrombosis and thrombocytopenia after receiving the first dose of the ChAdOx1 nCoV-19 adenoviral vector vaccine  and  Ad26.COV2.S  vaccine against coronavirus disease 2019 (Covid-19).

In a scientific report by Schultz NH, Sørvoll IH, Michelsen AE, Munthe LA, Lund-Johansen F, Ahlen MT, et al. Thrombosis and thrombocytopenia after ChAdOx1 nCoV-19 vaccination. N Engl J Med. 2021;384(22):2124–30, described 5 case reports of patients who presented with venous thrombosis and thrombocytopenia 7 to 10 days after receiving the first dose of the ChAdOx1 nCoV-19 adenoviral vector vaccine against coronavirus disease 2019 (Covid-19). Two patients of them died.

In turn of a scientific report by See I, Su JR, Lale A, Woo EJ, Guh AY, Shimabukuro TT, et al. US case reports of cerebral venous sinus thrombosis with thrombocytopenia after Ad26. COV2.S vaccination, March 2 to April 21, 2021. JAMA. 2021;325(24):2448–56. https://doi.org/10.1001/jama.2021.7517, described 12 case reports of patients with thrombosis (CVST) and  thrombocytopenia following Ad26.COV2.S vaccine receipt. Three patients of them died.

8. The study also did not look at the frequency of side effects in individuals who got placebos in clinical trials. This comparison would have added a valuable perspective for the readers since it would better gauge what was caused by the vaccinate versus the injection.

Response: In line with the valuable suggestion of the reviewer, the discussion of the paper was enriched with clinical data on the comparison of the incidence of adverse effects between the vaccinated group and the control group of the discussed CoVID-19 vaccines.

Reviewer 2 Report

Manuscript deals with the subject of pharmacovigilance of COVID-19 vaccines among healthcare workers in Poland.

I congratulate the Authors such well prepared and important manuscript, which is very important in our understanding of pandemic, protective influence and safety of vaccination.

Manuscript is well prepared. I have no remarks to the content.

My question is if subjects, who previously have confirmed SARS-Cov-2  infection have different symptoms after vaccination (more or less pronounced?) comparing to previously healthy colleagues. This, if possible should be mentioned in the manuscript.

Author Response

Reviewer 2

We greatly appreciate You to accept our paper in the present form.

General comment:

Manuscript deals with the subject of pharmacovigilance of COVID-19 vaccines among healthcare workers in Poland.

I congratulate the Authors such well prepared and important manuscript, which is very important in our understanding of pandemic, protective influence and safety of vaccination.

Manuscript is well prepared. I have no remarks to the content.

My question is if subjects, who previously have confirmed SARS-Cov-2  infection have different symptoms after vaccination (more or less pronounced?) comparing to previously healthy colleagues. This, if possible should be mentioned in the manuscript.

Response: Thank you for the valuable question. Our additional analysis of the impact of the confirmed SARS-CoV-2 infection at baseline on the number and intensity of side effects after vaccination showed that in the case of the respondents vaccinated against Covid-19 based on mRNA technology (Pfizer, Moderna), a relationship between the duration of side effects and the confirmed SARS-Cov-2 infection at baseline was demonstrated. In both study groups, the duration of side effects after receiving the Covid-19 vaccine was significantly longer in people with  confirmed SARS-Cov-2 virus infection at baseline than those who did not have previously confirmed infection (4.8± 3.0 vs. 4.0 ± 3.1, p=0.0364- Group 1; 8.3± 2.5 vs. 5.7 ± 3.4, p=0.0052- Group 2). However, no correlation was observed between the confirmed SARS-Cov-2 infection at baseline and the number and intensity of side effects following the administration of SARS-Cov-2 virus vaccines among all study groups (p>0.005). The obtain results are presented in the work in Table 4.  

Reviewer 3 Report

Lines 132-138, the recruitment process should be described with elaboration and details. How many were available? How many approached? How may declined? How they were accessed etc. It should be described in such a manner that the study can be repeated.

Lines 174-181, do you mean validity and reliability checking? please add the related score of the instrument after the pilot testing.

Lines 171-172, this is unclear how and on what platforms the questionnaires have been distributed.

How about missing data?

Results should be started with the details of the data collection as sample size, missing, withdrawal etc.

A graph in the methods section can help with understanding the process of your research. 

Author Response

Reviewer 3

We would like to thank for a great work done during thoroughly reviewing of our article.

Comments

  1. Lines 132-138, the recruitment process should be described with elaboration and details. How many were available? How many approached? How may declined? How they were accessed etc. It should be described in such a manner that the study can be repeated.

Response: Thank you for your valuable attention. The methodological part of the work has been enriched with detailed information on the process of recruiting respondents to the survey.

2. Lines 174-181, do you mean validity and reliability checking? please add the related score of the instrument after the pilot testing.

Response: A pre-test procedure on a representative sample of 150 subjects (representative sample: 50- doctors, 50- nurses and 50- pharmacists), was used as a suitable method for validation and for assessing the psychometric properties of questionnaire. Internal consistency of the safety profile of Covid-19 vaccines questionnaire is, the extent to which the items are correlated—was determined by calculating Cronbach’s α. Cronbach's α was 0.79, which proves about an excellent internal consistency.

3. Lines 171-172, this is unclear how and on what platforms the questionnaires have been distributed.

Response: The survey was implemented using the CAWI (Computer – Assisted Web Interview) method. Poznan University of Medical Sciences website and social media (e.g., Facebook, scientists association) were used to distribute questionnaires. Questionnaires were also distributed electronically (by e-mail) throughout the country.

4. How about missing data?

Response: The vast majority of the questions contained in the questionnaire were obligatory, which means that in order to be able to move on to the next question, the respondent had to answer the previous question first. The response rate, defined as the number of adequately completed online forms, was 86%. Only 100% correctly completed questionnaires were included in the study.

5. Results should be started with the details of the data collection as sample size, missing, withdrawal etc.

Response:  In line with the valuable suggestion of the reviewer, results part of the paper were enriched with the results of the respondents recruitment process.

6. A graph in the methods section can help with understanding the process of your research

Response: In accordance with the valuable suggestion of the reviewer, a graph (Figure 1) in the methods section was prepared showing the recruitment process and the research methods applied.

Round 2

Reviewer 3 Report

No more comments.